# Generation of Ultra-Thin-Shell Microcapsules Using Osmolarity-Controlled Swelling Method

**DOI:** 10.3390/mi11040444

**Published:** 2020-04-23

**Authors:** Jianhua Guo, Lihua Hou, Junpeng Hou, Jiali Yu, Qingming Hu

**Affiliations:** School of Mechatronics Engineering, Qiqihar University, Wenhua Street 42, Qiqihar 161006, Heilongjiang, China; guo1034@sina.com (J.G.); hlh0688@126.com (L.H.); houjunpengyjs@163.com (J.H.); futuredreams520@sina.com (J.Y.)

**Keywords:** microcapsules, double-emulsion drops, osmotic pressure, ultra-thin-shell, microfluidics

## Abstract

Microcapsules are attractive core-shell configurations for studies of controlled release, biomolecular sensing, artificial microbial environments, and spherical film buckling. However, the production of microcapsules with ultra-thin shells remains a challenge. Here we develop a simple and practical osmolarity-controlled swelling method for the mass production of monodisperse microcapsules with ultra-thin shells via water-in-oil-in-water (W/O/W) double-emulsion drops templating. The size and shell thickness of the double-emulsion drops are precisely tuned by changing the osmotic pressure between the inner cores and the suspending medium, indicating the practicability and effectiveness of this swelling method in tuning the shell thickness of double-emulsion drops and the resultant microcapsules. This method enables the production of microcapsules even with an ultra-thin shell less than hundreds of nanometers, which overcomes the difficulty in producing ultra-thin-shell microcapsules using the classic microfluidic emulsion technologies. In addition, the ultra-thin-shell microcapsules can maintain their intact spherical shape for up to 1 year without rupturing in our long-term observation. We believe that the osmolarity-controlled swelling method will be useful in generating ultra-thin-shell polydimethylsiloxane (PDMS) microcapsules for long-term encapsulation, and for thin film folding, buckling and rupturing investigation.

## 1. Introduction

A microcapsule is a micrometer-scale core-shell structure, with compartments encapsulated in a solid shell. This kind of core-shell configuration not only can protect the core materials from external disturbance or even contamination, but can also perform the on-demand delivery and release of the aqueous core under various external stimuli [1,2]. Thus, microcapsules can serve as model systems in various applications, such as drug delivery and controlled release [3,4,5,6,7], photonic capsule sensors [8,9,10], artificial microbial environments [11,12], foods [13,14], and biomolecular sensing [15,16].

Microfluidics technology supplies a simple and effective method for the production of various microcapsules with well-defined compositions and structures via double-emulsion drops templating [1,17]. For instance, by solidifying the liquid shell of the double-emulsion drops using temperature, pH, chemistry or light-sensitive materials, microcapsules with controlled release behaviors can be mass produced [18,19,20,21,22,23,24,25]. A porous shell endows microcapsules with a size-selective permeability [26]. Multicompartment microcapsules with multi-cores or capsule-in-capsule structures are designed for co-encapsulation and diverse programmable sequential release [27,28,29,30,31]. Smart microcapsules, which have magnetic materials embedded on the shell or in the cores, have been used for direction-specific delivery and release [32,33,34].

In particular, microcapsules with ultra-thin shells are critical templates for the above mentioned applications, and for studies of folding and buckling behaviors of spherical thin films [35]. However, the production of microcapsules with ultra-thin shells remains a challenge. It is very difficult to fabricate a double-emulsion drop with an ultra-thin shell less than several hundred nanometers using classic capillary microfluidic devices. There have been some configurations of capillary microfluidics to fabricate such ultra-thin-shell double-emulsion drops and microcapsules [36,37,38,39,40,41,42]. For example, an ultra-thin middle layer of double-emulsion drops can be stably created by squeezing a thin layer of middle fluid between the inner wall of the capillary and the innermost fluid [36]. The middle fluid needs to have high affinity to the capillary wall. The phase compositions and wettability of the middle fluid, however, are strictly required to create the thin shell. Therefore, developing an alternative method with simple configuration and practical performance for mass production of ultra-thin-shell microcapsules is of great significance.

In this study, we demonstrate an easy and practical method to generate highly monodisperse ultra-thin-shell microcapsules via osmolarity-controlled swelling behaviors of water-in-oil-in-water (W/O/W) double-emulsion drops. To achieve this, a classic capillary microfluidic device is firstly designed to fabricate monodisperse W/O/W double-emulsion drops, as shown in Figure 1a. Then, an osmolarity-controlled swelling method is developed to make the shell thinner by increasing the drop diameter, as the schematic illustration shown in Figure 1b. Here, the swelling process of the emulsion drops is simply controlled by the osmotic pressure between the inner aqueous and the suspending medium. The shell material, polydimethylsiloxane (PDMS), enables the transportation of water into or out of the drops when they are subjected to an osmotic pressure difference [43,44]. The shell thickness of the double-emulsion drop after swelling is figured out by measuring the volume of the PDMS shell phase after an electro-triggered rupturing, the value of which can be several hundreds of nanometers or less. Finally, the double-emulsion drops are transformed into ultra-thin-shell microcapsules by a thermal curing method to solidify the PDMS shell phase. This method enables the mass production of microcapsules with ultra-thin shells, and the shell thickness can be controlled precisely.

## 2. Materials and Methods

### 2.1. Materials

Polyvinyl alcohol (PVA, 87–89% hydrolyzed, average Mw = 13,000–23,000), potassium chloride (KCl), octadecyltrichlorosilane (OTS), and methylene blue were purchased from Sigma-Aldrich (St. Louis, MO, USA). PDMS kit (Sylgard 184) and silicone oil (50 cSt, PMX-200) were purchased from Dow Corning (Midland, MI, USA). As the outer continuous phase (W_outer_), aqueous solution of 5 wt% PVA was used. PVA served as a surfactant in this case. As the middle oil phase (O_middle_), a mixture of PDMS and silicone oil with the volume ratio of 3:1 was employed, in which 10 wt% curing agent was added for solidifying the double-emulsion drops into microcapsules. As the inner phase (W_inner_), aqueous solution of 1 wt% PVA was used. Depending on different experiments, KCl was added into the inner phase and suspending medium to adjust the osmotic pressure between the inner and outer of the PDMS shell. In some experiments, methylene blue was dissolved into the inner core as a dye for better visualization. All water used in this study was deionized (DI) water unless otherwise noted.

### 2.2. Fabrication of the Glass Capillary Microfluidic Device

A classic glass-capillary microfluidic device was used to fabricate the W/O/W double-emulsion drops, as described previously [17,45]. Briefly, two circular capillary tubes (ID of 0.58 mm, OD of 1.03 mm, World Pricision Instrument Inc., Sarasota, FL, USA) were given tapered openings of 38 and 170 μm in diameter using a micropipette puller (P-97, Sutter Instrument Inc., Novato, CA, USA) and a microforge (Narishige MF-900, Tokyo, Japan). The outside of the glass capillary tube for the inner fluid was hydrophobically functionalized with OTS to enhance the wettability of the capillary tube with oil phase, facilitating the fabrication of W/O/W double-emulsion drops. The two tapered circular capillary tubes were coaxially aligned and opposed to each other within a square glass capillary (ID of 1.05 mm), which were spaced from each other by 80 μm, as shown the optical microscope image in Figure 2a. All the capillaries and needles used for channels were connected using a transparent epoxy. Such a configuration possessed the hydrodynamic flow-focusing and coflowing functions for one-step fabrication of highly monodisperse W/O/W double-emulsion drops.

### 2.3. Generation of Monodisperse W/O/W Double-Emulsion Drops

To generate W/O/W double-emulsion drops, three liquid phases, W_outer_, O_middle_ and W_inner_, were pumped into the microfluidic device by three syringe pumps (Harvard Apparatus, PHD 2000 Series, Holliston, MA, USA). The generation process of the W/O/W double-emulsion drops in the device was monitored by a digital CMOS camera (Prime 95B, Qimaging, Surrey, BC, Canada). By tuning the flow rates of three phases, monodisperse W/O/W double-emulsion drops were formed through one-step emulsification, as shown in Figure 2a, where the flow rates of outer continuous phase (*Q*_outer_), middle oil phase (*Q*_middle_) and inner aqueous phases (*Q*_inner_) are 16,000, 120 and 1000 μL/h, respectively. The double-emulsion drops generated using this classic capillary microfluidic device were highly monodisperse, of which the size distribution can be maintained at 2.35% or less, as shown in Figure 2b and the inset bar graph. The drop diameter (*d*) was controlled precisely by varying *Q*_outer_. For example, as *Q*_outer_ increased from 8 to 18 mL/h, *d* decreasedfrom 159.9 to 125.4 μm, as shown in the graph in Fig. 2c, where *Q*_middle_ and *Q*_inner_ are maintained at 120 and 1000 μL/h, respectively. The reason can be understood that the increased *Q*_outer_ induces a higher shear force acting on the emulsion drop, leading to the decrease in *d*. 

In addition, the shell thickness (*h*) of double-emulsion drops was varied by modulating the flow rates ratios of *Q*_middle_/*Q*_inner_, as shown in the graph and inset images in Figure 3. The stars represent the experimental results of relative shell thickness *2h/d*, which could be varied from 0.041 up to 0.386 with excellent control and reproducibility. The inset optical microscope images show emulsion drops with varying relative shell thickness: 0.041, 0.128, 0.196, 0.232 and 0.386. Table 1 shows the W/O/W double-emulsion drops with varying relative shell thickness, diameter, shell thickness for the relevant flow ratios of *Q*_middle_/*Q*_inner_. The line in Figure 3 is plotted to illustrate the relative shell thickness *2h/d* as a function of *Q*_middle_/*Q*_inner_, which is rearranged by the mass balance equation in reference [16,45,46]. The experimental results are in good agreement with the line plotted in Figure 3, indicating that the microfluidic emulsion technology is good at controlling the shell thickness of the double-emulsion drops. However, it is difficult to generate monodisperse double-emulsion drops with ultra-thin shells using this classic microfluidic emulsion technology. Based on the reliable generation of monodisperse double-emulsion drops, the osmolarity-controlled swelling method was developed to produce the ultra-thin-shell double-emulsion drops and the resultant microcapsules.

### 2.4. Electrotriggered Rupturing of W/O/W Double-Emulsion Drops

In this study, the W/O/W double-emulsion drop needed to be triggered to rupture by applying an alternating current (AC) electric field to calculate the shell thickness. Two acupuncture needles immersed in the suspending medium acted as electrodes with a distance of 800 μm, through which a square-wave AC voltage signal was applied to develop an AC electric field between the electrodes. During this process, the AC signal was generated by a function generator (TGA 12104, TTi, Manchester, UK), and amplified by an amplifier (model 2350, TEGAM, Geneva, OH, USA). Under an appropriate field strength and frequency (30 V, 5 KHz) applied, the double-emulsion drop between the two electrodes ruptured immediately due to Maxwell−Wagner interfacial polarization [47,48,49].

## 3. Results and Discussion

### 3.1. Osmolarity-Controlled Swelling Behavior

The core diameter and shell thickness of the W/O/W double-emulsion drops can be tuned by the osmolarity-controlled swelling behavior. When the salt concentration of outer suspending medium is higher than it in the inner aqueous core, water in the suspending medium has a higher chemical potential than that of the inner core, as shown in the schematic illustration in Figure 4a. As a result, water in the suspending medium will penetrate through the oil shell into the core, leading to the osmolality-controlled swelling behaviors of the inner core [43,50]. Along with the swelling of the inner core, the oil shell becomes sufficiently thinner and thinner, leading to the generation of ultra-thin-shell double-emulsion drops.

To study the effect of osmolality-controlled swelling behavior on shell thickness of double-emulsion drops, the swelling phenomena of emulsion drops with varying osmotic pressure between the inner core and outer suspending medium are investigated. Firstly, double-emulsion drops with a given KCl concentration of 0.25 mol/L as the inner aqueous cores are generated in our capillary device, which are collected in 1 wt% PVA solution with KCl concentration of 0.25 mol/L to prevent the spontaneous swelling and coalescence of these drops, as shown in Figure 4b. Here, *Q*_outer_, *Q*_middle_ and *Q*_inner_ are 16,000, 120 and 1000 μL/h, respectively. The drop diameter is about 129.5 μm. According to the relationship between the relative shell thickness *2h/d* and *Q*_middle_/*Q*_inner_ shown in Figure 3, we can derive the shell thickness of the double-emulsion drop is 2.4 μm. Then, the KCl concentration of the suspending medium is double diluted by adding 1 wt% PVA solution every 30 min. Here, the dilution of the KCl concentration in the suspending medium induces the varying osmotic pressure between inner cores and the suspending medium, resulting in the osmolality-controlled swelling of the inner core. Meanwhile, the drop size is measured before every dilution process, as shown in the optical images in Figure 4b–d. After five times dilution, finally, the diameter of the double-emulsion drops increases from 129.5 to 360 μm along with the osmolality-controlled swelling process. Knowing the initial diameter, shell thickness and the swelled diameter of the double-emulsion, the shell thickness of the swelled emulsion can be derived geometrically, the value of which is 300 nm. As a result, the shell thickness of the emulsion decreases from 2.4 μm to about 300 nm, indicating that the osmolarity-controlled swelling method enables the mass production of microcapsules with ultra-thin-shell.

### 3.2. Measuring of the Shell Thickness

To evaluate the shell thickness of the double-emulsion drop after swelling, scanning electron microscope (SEM) imaging is the most direct way to get this value [8,36,40]. However, it needs expensive SEM equipment and multiple operations to take SEM images. In other way, if we know the initial data (diameter and shell thickness before swelling) and the swelled diameter of the double-emulsion drop, the shell thickness of the swelled emulsion drop can be derived geometrically, as demonstrated in Section 3.1. In some cases, however, the initial data of the original capsules are missing or not attainable, leading to the difficulty in calculating shell thickness. Here, we present a simple and useful strategy to quickly derive the shell thickness by measuring the volume of oil shell phase after the emulsion drop rupturing. As a comparison, the original emulsion drops are generated using the same flow rates with Section 3.1, the diameter and shell thickness of which are 129.5 and 2.4 μm, respectively. Time-lapse microscopy images in Figure 5 show the rupture process. Methylene blue is dissolved into the inner core as a dye for better visualization. When the emulsion drop is swelling to about 360 μm in diameter, the drop is triggered to rupture by applying an alternating electric field as described in Section 2.4. From 0.5 s, the aqueous core of the emulsion drop sprays out because of the interfacial tension, and the oil shell shrinks into a wrinkled membrane. At about 35 s, most of the inner phase is discharged from the core. The empty oil shell assembles into a spherical drop under the action of surface tension. After 75 s, the oil shell transformed into a tiny oil drop with a diameter of 64 μm. By comparing the diameters of the oil drop after rupturing and the double-emulsion drop before rupturing, we can readily figure out the shell thickness of the swelled double-emulsion drop, which is about 338 nm. This indicates that the shell thickness evaluated using the drop rupturing method (338 nm) is in good agreement with that derived geometrically (300 nm).

### 3.3. Curing Process – Transform the Double-Emulsion Drops to Solid Core-Shell Microcapsules

For some situations, the emulsion drops need to be transferred to microcapsules for long-term observation and thin shell buckling investigation [35]. Herein, we cure the PDMS shell at 37 °C for 24 h to obtain the microcapsules with highly robust ultra-thin shells, as the schematic illustration shows in Figure 6a. An number of solid microcapsules with an ultra-thin shell after thermal curing is shown in Figure 6b. As can be seen in Figure 2b and Figure 6b, the size distribution of the double-emulsion drops (polydispersity of 2.35%) and swelling produced ultra-thin-shell microcapsules (polydispersity of 2.56%) shows good uniformity, clearly indicating that the osmolality-controlled swelling method for mass production of ultra-thin-shell microcapsules is feasible and efficient. Furthermore, the solid ultra-thin-shell microcapsules can maintain their intact spherical shape for up to 1 year with no rupture in our long-term observation, indicating that the ultra-thin-shell PDMS microcapsules are robust as vesicles for long-term encapsulation and observation.

To demonstrate the uniformity of the shell thickness, we compress the ultra-thin-shell microcapsules by imposing a strong osmotic pressure. When these ultra-thin-shell microcapsules are placed in a suspending medium with KCl concentration of 0.5 mol/L, the inner aqueous phase diffuses out from the PDMS microcapsules, resulting in a volume shrinkage of the microcapsules. As the shrink process goes on, wrinkles develop on the ultra-thin-shell microcapsules. After 30 min, wrinkles with remarkable uniformity grow up on the shell of microcapsule, as shown in Figure 6c. Direct observation of uniform wrinkling on the surface of the ultra-thin-shell microcapsules after osmolality-controlled shrink confirms that the shell thickness of the microcapsules is ultra-thin and relatively uniform. We believe this kind of ultra-thin-shell PDMS microcapsule is an attractive template for studies of folding, buckling and rupturing behaviors of spherical thin films.

## 4. Conclusions

In this study, an osmolarity-controlled swelling method is presented to generate highly monodisperse microcapsules with an ultra-thin shell via W/O/W double-emulsion drops templating. The shell thickness of the emulsion drops can be tuned precisely off-chip through osmolarity-induced swelling of the emulsion drops. Here, this swelling process is simply controlled by the osmotic pressure between the inner aqueous and the suspending medium. Using this method, a shell thickness of hundreds of nanometers can be developed. Furthermore, after a thermal-curing process, the emulsion drops can be transformed to ultra-thin-shell microcapsules, which can maintain their intact spherical shape for up to 1 year with no rupture in our long-term observation. This method enables the mass production of microcapsules with ultra-thin shells, which are robust as vesicles for long-term encapsulation and observation, and attractive templates for studies of folding, buckling and rupturing behaviors of spherical thin films.

## Figures and Tables

**Figure 1 micromachines-11-00444-f001:**
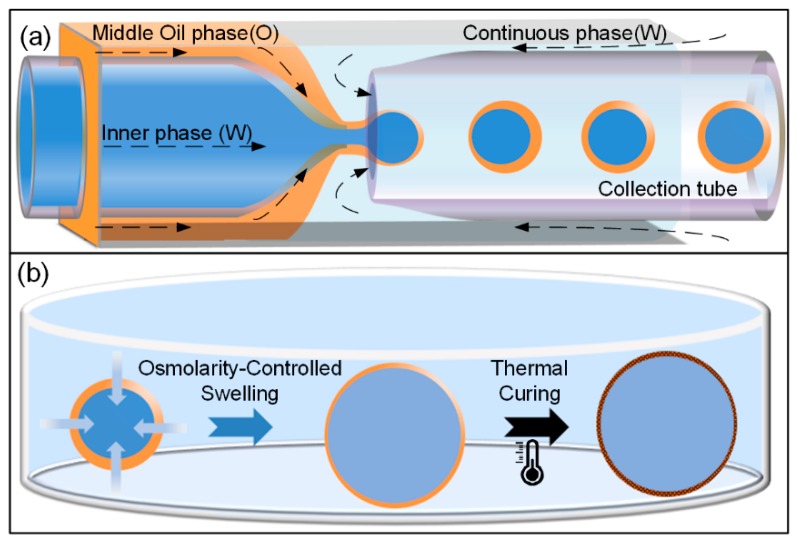
(**a**) Schematic illustration of the capillary microfluidic device for generating water-in-oil-in-water (W/O/W) double-emulsion drops; (**b**) schematic illustrations showing the osmolarity-controlled swelling process of W/O/W double-emulsion drop, and the thermal curing process from liquid emulsion drop to solid microcapsule.

**Figure 2 micromachines-11-00444-f002:**
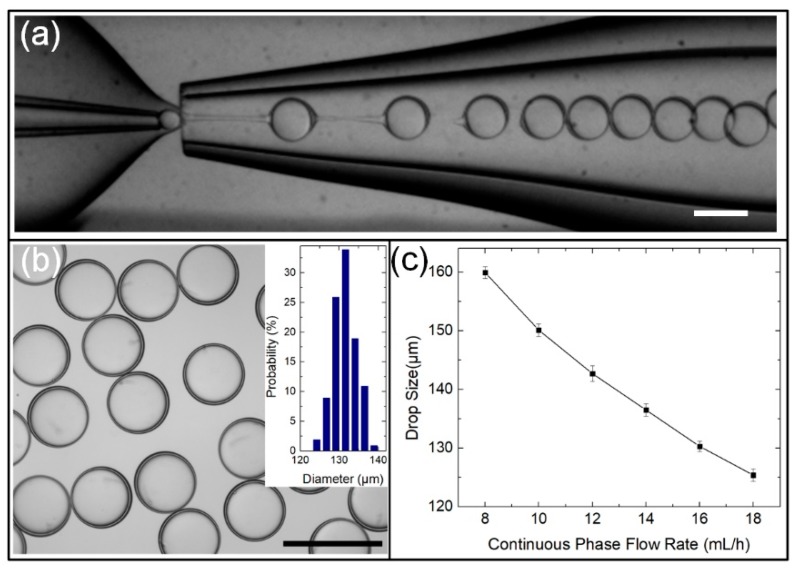
(**a**) Optical microscope image showing the inner configuration of the classic capillary microfluidic device and the generation of W/O/W double-emulsion drops using this device; (**b**) optical microscopy image of highly monodisperse W/O/W double-emulsion drops (inset: size distribution of the emulsion drops). (**c**) Influence of continuous phase flow rate on dimensions of double-emulsion drops, where *Q*_outer_ is varied from 8 to 18 mL/h; *Q*_middle_ and *Q*_inner_ are maintained at 120 and 1000 μL/h. Scale bars are 200 μm.

**Figure 3 micromachines-11-00444-f003:**
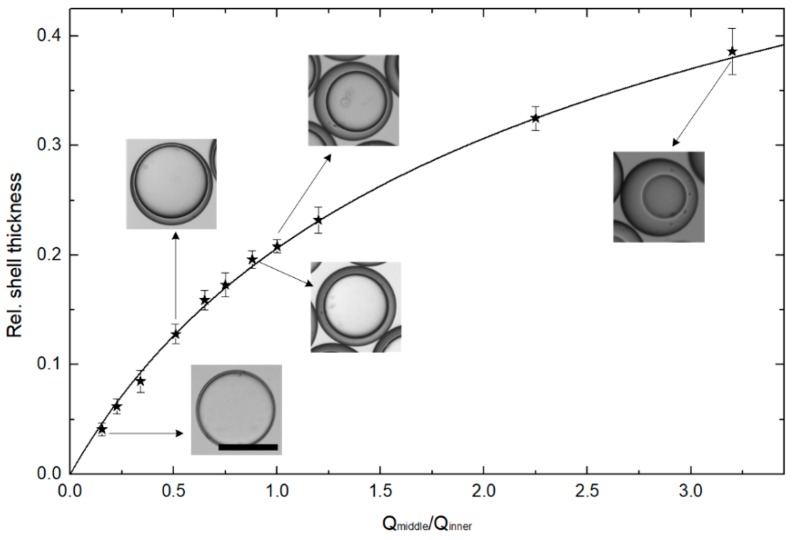
Experimental and theoretic shell thickness as a function of the flow rates ratios of the middle oil and inner aqueous phases. The inset optical microscope images showing emulsion drops with varying relative shell thickness: 0.041, 0.128, 0.196, 0.232 and 0.386. Scale bars are 100 μm.

**Figure 4 micromachines-11-00444-f004:**
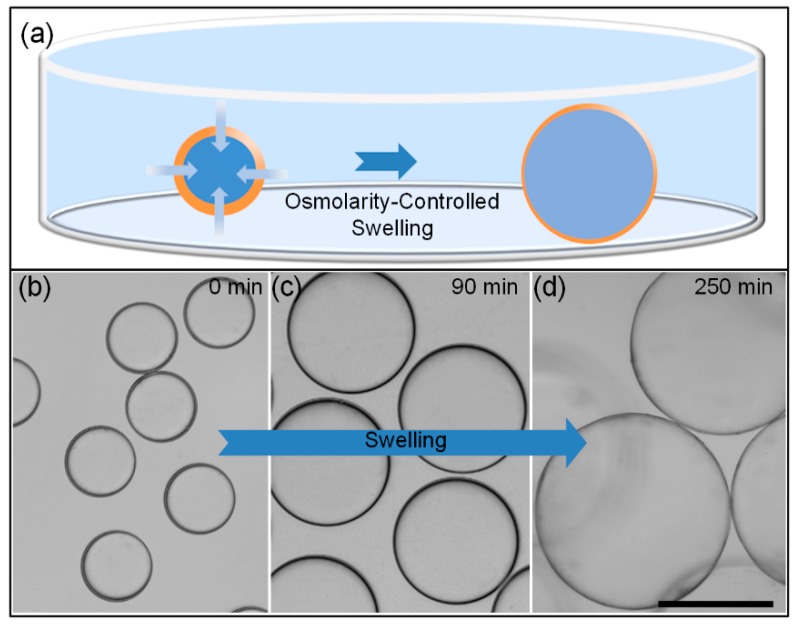
(**a**) Schematic illustration showing the osmolarity-controlled swelling process of the W/O/W double-emulsion drops; (**b**–**d**) Optical microscope images showing the osmolarity-controlled swelling process of W/O/W double-emulsion. Scale bars are 200 μm.

**Figure 5 micromachines-11-00444-f005:**
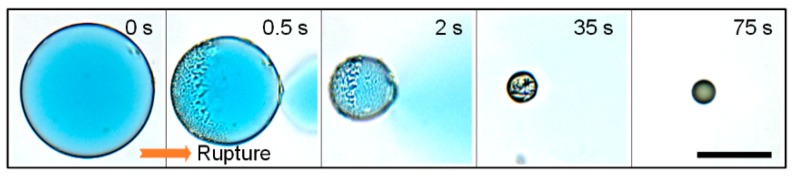
Time-lapse microscopy images showing the rupture process of the W/O/W double-emulsion drop. A digital charge coupled device (CCD) camera (Retiga 2000R, Qimaging) is used to take these images. Scale bar is 200 μm.

**Figure 6 micromachines-11-00444-f006:**
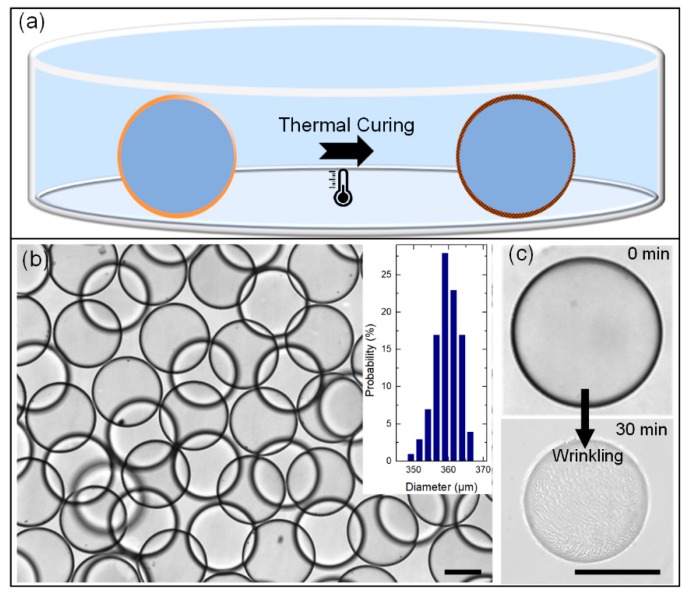
(**a**) Schematic illustration showing the liquid emulsion drop transformed to solid microcapsule via the thermal curing process; (**b**) solid microcapsules with ultra-thin shell after the thermal curing process (inset: size distribution of the microcapsules); (**c**) wrinkling of cured microcapsules by imposing a strong osmotic pressure. Scale bars are 200 μm.

**Table 1 micromachines-11-00444-t001:** W/O/W double-emulsion drops with varying relative shell thickness, diameter, shell thickness for the relevant flow ratios of *Q*_middle_/*Q*_inner_.

*Q*_middle_/*Q*_inner_	Rel. Shell Thickness	*D* (μm)	*h* (μm)
0.154	0.041	132.5	2.7
0.511	0.128	139.6	8.9
0.881	0.196	136.6	13.4
1.00	0.208	133.5	13.9
3.20	0.386	132.0	25.5

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
