# Peer review of "Generation of Ultra-Thin-Shell Microcapsules Using Osmolarity-Controlled Swelling Method"

_micromachines, 2020, doi:10.3390/mi11040444_

Round 1

Reviewer 1 Report

The paper describes an alternative method for producing ultrathin shell microcapsules using PDMS and silicone oil mixture. This method considered the production of microcapsules using a well-known microfluidic method, which are then expanded by osmotic pressure, leading to a shell thinning. Even if the proposed method is interesting, there are some important aspects that must be faced before considering the paper for the publication.

1) line 47-50. Authors describe the references 35-39 as the only methods for producing ultrathin microcapsules using microfluidics; however, there are others in the literature (some of them already cited in the paper), for example: doi.org/10.1002/cphc.201402606, doi.org/10.1002/admt.201800006, doi.org/10.1021/acs.langmuir.5b01129. The advantages of the presented methods compare with these papers must be also introduced in the paper.

2) line 59-61. The shell thickness is evaluated after the capsule rupture. If this is a commonly used strategy, references should be indicated in the text.

A part from that, since the goal is to derive the shell thickness after the swelling, starting from a known initial thickness (Figure 3), why this is not simply derived geometrically? In other words, knowing the initial thickness and the total diameter of the vesicles, the shell volume can be calculated; after swelling, this volume must be the same and thus, by knowing the final total diameter of the capsules, the shell thickness can be derived.

In literature, SEM or TEM images are also employed to evaluate the shell thickness; do they represent a possible investigation in this work as well?

3) line 73-74. A mixture of PDMS and silicone oil at 50cSt is used for the shell phase. I supposed that this is done to reduce the viscosity of the PDMS, supporting the capsules formation. Are there specific restrictions regarding the liquid phase that can be employed in the microfluidic device in order to produce the microcapsules? Since this is a technical paper, this should be investigated and described.
By further reducing the viscosity of the oil phase, would it be possible to directly obtain microcapsules with thinner shells?

4) Line 103. Please indicate brand and model of the syringe pump used; this can be important to understand how precise and stable need to be the applied flows.

5)  line 122-123. “are in good agreement with the linear relation plotted in Figure 3”. Figure 3 does not show a linear trend. Text correction is highly recommended.

6) The shell thickness “h” of the data in Figure 3 is not reported anywhere; while, it is the relative values. Since the paper aim to produce an ultra-thin shell, a would report the real values of “h” as well. For example, authors could add a second Y-axis on the graph of Figure 3.

7) line 155. “is extremely thin”. Actually, from Figure 4 it is difficult to notice the difference of the thickness before and after the swelling, therefore I suggest to the authors to report same data on the text.

8) paragraph 3.2. As mention in comment 2, the shell thickness produced in this way can be geometrically calculated. Since authors do not propose a direct measure of the shell (for example by electron microscopy), I suggest to try this indirect method as well and compare the results.

9) Figure 5. Authors says that they add some methylene blue in the inside solution of the capsule to increase the contrast; this is a good idea. However, since the images reported in Figure 5 are with colors, it is not clear to me if they are snapshots taken directly with the camera or if they have been post-processed. Notably, the images look completely different than those that are presented in Figures 2,3,4. Authors must indicate in the text (or in the caption) if the imaged have been post-processed and be sure that this did not alter their quantification.

10) line 212. “few hundreds of nanometers or even less”. The only thickness report in the paper is 338nm. Therefore, it is not showed that it is possible to go less than 100 nm. Conclusion must be rephrased.

Reviewer 2 Report

In their manuscript, the authors show a strategy for the production of microcapsules with ultra-thin shells. They take advantage of the swelling behavior of microcapsules exposed to different osmotic pressures. This idea is smart and the way they performed the experiments is well done and fully explained. The English is in a few parts is a bit hard to read. I think some editing on the text is required.

Overall, the manuscript shows a nice piece of scientific work and has clear value for readers from various scientific fields. I therefore can recommend publishing this work in Micromachines.

Author Response

We thank the reviewer for the positive comment. To make the manuscript easy to read, the whole manuscript is re-edited by a native English speaker.

Reviewer 3 Report

In this paper, authors explore the formation of PDMS-based core-shell capsules through the use of microfluidic double emulsion techniques. Authors present a microfluidic approach for controlling the inner and middle phase diameter by modulating the flow rates used on-chip and control the shell thickness through osmotic inhomogeneity between the inner and outer phases. In addition to direct measurements of PDMS shell thickness achieved by this approach, authors present increased stability and control over storage and release of molecules encapsulated. The results of this work further extend previous published data related to PDMS core-shell structure and offer a robust approach for tuning shell thickness.

While the manuscript presents a very interesting approach for production of core-shell structures and tuning the resulting capsule’s size and thickness by osmotic pressure, additional experiments along with revision of the text are required, as several key details are missing. The suggested further experiments would highly increase the impact of this work on the readership of Micromachines.

  1. Authors discuss the relevance of external stimuli on cargo release in the introduction, yet they do not give any examples of such approaches and how the presented technique differs from those published by, e.g. the Weitz lab. This needs to be extended and clarified.
  2. Authors claim that specific surface wetting treatment is required for the formation of double emulsion, yet there is no mention of the specific approach taken to deal with this issue. It would be ideal to include a discussion on the merit of capillary microfluidics vs. PDMS devices, and specifically the use of 3D PDMS devices such as has recently been published by Toprakcioglu et al. Biomacromolecules 18 (11), 3642-3651 and Barea et al. Micromachines 2019, 10 (6) , 412.
  3. Selection of the composition of the different phases should be discussed. In particular, the use of PVA is not explained at any point. Would this serve as a surfactant in this case and if not, did authors use any other surfactants. Similarly, why was KCl selected?
  4. Flow rates described in section 2.3 seem to be extremely high, reaching up to 18 mL/h while commonly these are kept to the range of several hundred uL/h. Why were these flow rates selected and would the formation of stable double emulsions be feasible using lower flow regimes? Furthermore, the distribution of low rates seems a bit arbitrary. Additional discussion might clarify this issue.
  5. Shell thicknesses presented in Fig. 3 are only described in relative values. Authors should provide a summary table shell thickness in um or nm for the relevant flow rates.
  6. In section 3.2, authors present the use of alternating electrical field to induce capsule rupturing. This is not presented in the Materials and Methods section and is very confusing. Authors should present this technique and specify the parameters used.
  7. In the same section, authors claim ‘From 0.5 s, the aqueous core of the emulsion drop sprays out because of the interfacial tension, and the oil shell shrinks into a wrinkled membrane.’ This is highly confusing as if this is still a liquid oil shell no wrinkling should appear. This means that the PDMS layer has already polymerized without any curing. This needs to be further investigated as it leads this reviewer to believe that there are crucial data still missing. It would further be interesting if authors compare these results with those described in Yin et al. Sci Rep. 2014;4:5710.
  8. Authors do not seem top de-emulsify the formed PDMS capsules further to curing and thus do not remove any excess oil. It is thus unclear whether the stability of formed capsules and their ability to prevent the release of dye molecules is governed by the remaining oil layer of by the PDMS shell. As PDMS by itself is know to be quite porous and should not limit diffusion of aqueous solution over long periods of time, I suspect that the presented results need to be compared to those of PDMS capsules in which the oil has been completely removed.
  9. Further to the above comment, authors should include data about methylene blue release kinetics following osmotic pressure as shown in Fig. 6c. Additional discussion regarding encapsulation of different molecular species and release kinetics is strongly encouraged by including further references, e.g., Elbers et al. Chem. Mater. 2015, 27, 5, 1709-1719 and Teixeira et al. Macromolecules 2014, 47, 23, 8231-8237.
  10. The manuscript requires an extensive text revision as it is rife with typos and/or very difficult to understand, especially in the abstract, introduction and discussion. Such examples are: Ultra-shin-shell instead of ultra-thin-shells (line 18), sample instead of simple (line 34) and the use of the term ‘sphericity’ out of context.

Round 2

Reviewer 1 Report

The revised version of the paper is now suitable for publication

Author Response

We thank the reviewer for the positive comment.

Reviewer 3 Report

Authors have made a good effort in responding to the previous comments, yet some issues have not been resolved thoroughly, as previously suggested:

  1. Authors claim that specific surface wetting treatment is required for the formation of double emulsion, yet there is no mention of the specific approach taken to deal with this issue. 
  2. Flow rates described in section 2.3 seem to be extremely high, reaching up to 18 mL/h while commonly these are kept to the range of several hundred uL/h. Why were these flow rates selected and would the formation of stable double emulsions be feasible using lower flow regimes? 
  3. In section 3.2, authors claim ‘From 0.5 s, the aqueous core of the emulsion drop sprays out because of the interfacial tension, and the oil shell shrinks into a wrinkled membrane.’ This is highly confusing as if this is still a liquid oil shell no wrinkling should appear. This means that the PDMS layer has already polymerized without any curing. This needs to be further investigated as it leads this reviewer to believe that there are crucial data still missing. It would further be interesting if authors compare these results with those described in Yin et al. Sci Rep. 2014;4:5710.
  4. Authors do not seem top de-emulsify the formed PDMS capsules further to curing and thus do not remove any excess oil. It is thus unclear whether the stability of formed capsules and their ability to prevent the release of dye molecules is governed by the remaining oil layer of by the PDMS shell. As PDMS by itself is know to be quite porous and should not limit diffusion of aqueous solution over long periods of time, I suspect that the presented results need to be compared to those of PDMS capsules in which the oil has been completely removed.
  5. Further to the above comment, authors should include data about methylene blue release kinetics following osmotic pressure as shown in Fig. 6c. Additional discussion regarding encapsulation of different molecular species and release kinetics is strongly encouraged by including further references, e.g., Elbers et al. Chem. Mater. 2015, 27, 5, 1709-1719 and Teixeira et al. Macromolecules 2014, 47, 23, 8231-8237.

In addition to these comments, some of the revised text remains unclear:

  1. Line 190 ‘In some cases, however, we don’t know the initial data of the original capsules.’ Authors should give a detailed analysis of what this data is and why is it not attainable in the approach taken.
  2. Line 203-204 ‘This shell thickness evaluated using the drop rupturing 203 method is in good agreement with it derived geometrically.’ This sentence is highly unclear.
  3. Authors would benefit from revising the text to avoid additional spelling mistakes such as ‘ultra-shin-shell’ on line 19 (which has previously been pointed), line 243 ‘highly uniformity’ etc.
